# Transcriptomic Analysis of Hub Genes Reveals Associated Inflammatory Pathways in Estrogen-Dependent Gynecological Diseases

**DOI:** 10.3390/biology13060397

**Published:** 2024-05-30

**Authors:** Elaine C. Pasamba, Marco A. Orda, Brian Harvey Avanceña Villanueva, Po-Wei Tsai, Lemmuel L. Tayo

**Affiliations:** 1School of Graduate Studies, Mapúa University, Manila City 1002, Philippines; elcunanan@mymail.mapua.edu.ph (E.C.P.); maorda@mymail.mapua.edu.ph (M.A.O.); bhavillanueva@mymail.mapua.edu.ph (B.H.A.V.); 2School of Chemical, Biological, and Materials Engineering and Sciences, Mapúa University, Manila City 1002, Philippines; 3Department of Food Science, National Taiwan Ocean University, Keelung 20224, Taiwan; powei@mail.ntou.edu.tw; 4Department of Biology, School of Health Sciences, Mapúa University, Makati City 1203, Philippines

**Keywords:** endometriosis, gynecological cancers, WGCNA, inflammation, drug repurposing

## Abstract

**Simple Summary:**

Gynecological diseases still make up a large percentage of the overall global disease burden. While oral contraceptives and gonadotropin-releasing hormone drugs for endometriosis and gynecological cancers exist, their known inflammatory side effects can counteract progress in therapy. With this, the present study made use of a systems biology approach to identify correlations between gynecological diseases using gene expression data from DNA microarray samples that contain endometriosis, ovarian cancer, cervical cancer, and endometrial cancer. The highly preserved gene modules and their top interacting hub genes were determined to provide a further understanding of the signaling pathways and biological processes affected. Potential drugs were screened based on the upregulated and downregulated hub genes, which identified drug candidates that have known anti-inflammatory effects, implying the potential of specific inflammatory pathways in estrogen-dependent gynecological diseases as a therapeutic avenue.

**Abstract:**

Gynecological diseases are triggered by aberrant molecular pathways that alter gene expression, hormonal balance, and cellular signaling pathways, which may lead to long-term physiological consequences. This study was able to identify highly preserved modules and key hub genes that are mainly associated with gynecological diseases, represented by endometriosis (EM), ovarian cancer (OC), cervical cancer (CC), and endometrial cancer (EC), through the weighted gene co-expression network analysis (WGCNA) of microarray datasets sourced from the Gene Expression Omnibus (GEO) database. Five highly preserved modules were observed across the EM (GSE51981), OC (GSE63885), CC (GSE63514), and EC (GSE17025) datasets. The functional annotation and pathway enrichment analysis revealed that the highly preserved modules were heavily involved in several inflammatory pathways that are associated with transcription dysregulation, such as NF-kB signaling, JAK-STAT signaling, MAPK-ERK signaling, and mTOR signaling pathways. Furthermore, the results also include pathways that are relevant in gynecological disease prognosis through viral infections. Mutations in the *ESR1* gene that encodes for ERα, which were shown to also affect signaling pathways involved in inflammation, further indicate its importance in gynecological disease prognosis. Potential drugs were screened through the Drug Repurposing Encyclopedia (DRE) based on the up-and downregulated hub genes, wherein a bacterial ribosomal subunit inhibitor and a benzodiazepine receptor agonist were the top candidates. Other drug candidates include a dihydrofolate reductase inhibitor, glucocorticoid receptor agonists, cholinergic receptor agonists, selective serotonin reuptake inhibitors, sterol demethylase inhibitors, a bacterial antifolate, and serotonin receptor antagonist drugs which have known anti-inflammatory effects, demonstrating that the gene network highlights specific inflammatory pathways as a therapeutic avenue in designing drug candidates for gynecological diseases.

## 1. Introduction

Currently, gynecological diseases still make up approximately 4.5% of the overall global disease burden, even surpassing major health concerns, such as malaria, ischemic heart disease, tuberculosis, and maternal conditions [1]. These diseases pertain to conditions related to the dysfunction of the female reproductive system. One of the most common gynecological diseases is endometriosis (EM). It is a condition in which endometrium grows outside the uterus, preventing the blood and tissue it releases during the menstrual cycle from exiting the body. It leads to inflammation, scarring, and abnormal tissue growth that can be a factor in increased risk for gynecological cancer [2]. Particularly, chronic inflammation is a major hallmark of cancer, and it is often experienced by patients suffering from the most common gynecological cancers—endometrial cancer (EC), ovarian cancer (OC), and cervical cancer (CC) [3], making it considered a biological mechanism that underpins carcinogenesis due to its ability to induce excessive production of pro-inflammatory cytokines which promote cellular proliferation and reduce apoptosis that contributes to tumor growth. Obesity and metabolic syndromes are well-established risk factors for endometrial and ovarian cancers due to visceral fats and adipokines promoting a pro-inflammatory state within the microenvironment [4,5,6]. On the other hand, chronic inflammation of the cervical epithelium is a result of human papillomavirus (HPV) infection and often leads to cervical carcinogenesis and progression [7].

Given that gynecological diseases are estrogen-dependent conditions, oral contraceptives and gonadotropin-releasing hormone (GnRH) agonists that maintain steady hormone levels are commonly given as medications. However, reports have shown that oral contraceptive use by women with polycystic ovarian syndrome (PCOS) increases the risk of inflammatory and coagulatory disorders, while aberrant GnRH secretion may even cause inflammation-induced ovarian dysfunction [7,8]. Concerning this, estrogen plays a major role in regulating immune and inflammatory responses, further making it plausible to investigate the molecular underpinnings of inflammation and estrogen production in gynecological diseases. This can be made possible through weighted gene co-expression network analysis (WGCNA) [9,10], which is a systems biology approach that characterizes the gene associations in the gene expression data of several diseases to identify significant gene clusters (modules) and elucidate the key hub genes and main pathways that are dysregulated. The identification of upregulated and downregulated hub genes further allows for the scanning of potential already-existing drugs that can induce countering effects based on the provided gene signature while determining significant pathways opens therapeutic opportunities. Several studies making use of WGCNA to explore gynecological cancers have further expanded our understanding, such as uncovering a new prognostic genetic marker for cervical cancer and identifying important pathways in endometrial cancer tumorigenesis [11,12].

With the growing evidence of significant associations between endometriosis and gynecologic cancers [13], this study aims to further investigate the role of inflammation in the prognosis of gynecological diseases using the gene expression dataset of EM (GSE51981), EC (GSE17025), OC (GSE63885), and CC (GSE63514) that were sourced from the Gene Expression Omnibus (GEO). Through WGCNA, five highly preserved modules were identified across all the datasets, which were then found to be strongly involved in transcription regulation, protein binding, and cell cycle via functional annotation and pathway enrichment analysis. The construction of a protein–protein interaction network revealed the key hub genes within each module that allowed the screening of potential repurposed drugs.

## 2. Materials and Methods

### 2.1. Dataset Acquisition and Pre-Processing

Through National Center for Biotechnology Information: Gene Expression Omnibus (NCBI GEO) [https://www.ncbi.nlm.nih.gov/geo/ accessed on 30 October 2023], DNA microarray datasets containing expression data of endometrial tissue and tumor samples from patients who have endometriosis (EM), ovarian cancer (OC), cervical cancer (CC) and endometrial cancer (EC). The sampling of this study was limited to the datasets carried out through GPL570-HG-U133 Plus 2 Affymetrix Human Genome U133 Plus 2.0 Array to prevent inconsistencies caused by differences in probe design or normalization [14,15]. Overall, a total of 77 samples were for EM, 101 samples were used for OC, 104 samples for CC, and 91 samples for EC. Table 1 presents the summarized information of the datasets used for the analysis.

Through the getGEO function of Bioconductor in RStudio (http://www.bioconductor.org accessed on 30 October 2023), microarray datasets were extracted and underwent quantile normalization using robust multi-array average (RMA). The phenodata that contain information on the control samples were used to remove normal cell expression data to retain only the samples from endometrial and cancer tissues. Then, expression values were log_2_ transformed, and genes without expression values were removed using the goodSamplesGenes function, as well as outliers in the sample clustering dendrograms. After log_2_ transformation, gene expression values falling below the minRelativeWeight threshold were denoted as missing entries in the data and removed. This is performed to achieve a stable list of good samples and genes and avoid errors in the calculation [16]. The expression data were then filtered to remove genes with expression values lower than the 20% percentile cut-off in the mean and variance. All genes that were consistently common across datasets were used for the subsequent analysis.

### 2.2. Weighted Gene Co-Expression Network Analysis (WGCNA)

#### 2.2.1. Scale-Free Network

In approximating the scale-free network, the pickSoftThreshold function of the WGCNA R [17] package (v4.3.1.) was first used to plot the scale-free topology fit versus power index (1–20) to estimate the appropriate soft-thresholding power (β), which is the lowest power where the scale-free topology criterion is met. This is when the distribution of the number of gene connections follows the power-law distribution, and it is usually evaluated by plotting the scale-free topology fit versus soft-thresholding power to measure the relativity of the co-expression network to the linear relationship between log of connectivity and log of connectivity probability. The power at which the scale-free topology fit reaches a plateau often indicates a good fit [18]. Hence, the softConnectivity function was used to plot the linear model fitting the *R*^2^ index to quantify the goodness-of-fit of the expression data of each dataset at the chosen soft-thresholding power. The dataset exhibiting the best fit is selected as the reference dataset for the WGCNA.

#### 2.2.2. Network Construction and Module Identification

After selecting the reference dataset, it is transformed into adjacency matrices calculated using Pearson’s correlation for network construction. Then, the adjacency matrices were converted into Topological Overlap Matrices (TOM) using the TOMsimilarity function to calculate the similarities between genes for clustering. The hclust function was used to conduct hierarchal clustering and produce gene dendrograms using highly correlated genes. Module identification and generation of TOM dissimilarity dendrogram were performed through the cutreeDynamic function, where the deep split parameter was set to 2 (medium sensitivity) and the minimum cluster size was set to 30. Lastly, eigen correlation was performed to merge the modules with similar expression profiles to produce more significant results.

#### 2.2.3. Module Preservation Analysis

Using the module preservation function, the gene co-expression network preservation of EM, OC, CC, and EC modules was analyzed with the network type “signed” and the number of permutations set to 100. This is to measure the degree to which the connectivity patterns of the reference network align with the other datasets to evaluate the biological relevance of the gene modules identified.

### 2.3. Gene Ontology (GO) and Kyoto Encyclopedia of Genes and Genomes (KEGG) Pathways

Genes in each highly preserved module were extracted from RStudio and sent to the Database for Annotation, Visualization, and Integrated Discovery (DAVID) web-based tool for functional annotation and pathway enrichment analysis to understand the biological foundations of the identified modules. Prior to analysis, the Affymetrix gene IDs resulting from WGCNA were converted into Entrez IDs for database identification. Functional annotation was performed via the Gene Ontology (GO) database to investigate the Biological Processes (BP), Molecular Functions (MF), and Cellular Components (CC) involved. Furthermore, the Kyoto Encyclopedia of Genes and Genomes (KEGG) database was used to provide information on the biological pathways that are affected by the modules. The classification stringency was set to “medium,” and GO Terms and KEGG Pathways with *p*-values lower than 0.05 were considered statistically significant in the analysis.

### 2.4. Protein-Protein Interaction (PPI) and Hub Genes

The Entrez IDs of the genes in each highly preserved module were then sent to the Search Tool for the Retrieval of Interacting Genes/Proteins (STRING) database to identify genes with potential protein-protein interaction (PPI) networks. Afterward, the networks constructed were exported to Cytoscape, and through the CytoHubba feature, the top 10 genes with the highest interactions within their corresponding modules were identified by ranking them based on degree centrality, classifying them as hub genes.

### 2.5. Signature-Based Approach for Drug Repurposing

Before scanning for potential drugs, the identified hub genes were first grouped into upregulated and downregulated hub genes through differential expression analysis in GEO2R [https://www.ncbi.nlm.nih.gov/geo/geo2r/ accessed on 20 November 2023]. Then, each group of genes was exported to the Drug Repurposing Encyclopedia (DRE) to search from the Molecular Signatures Database (MSigDB) and Connectivity Map (CMap) for already-existing drugs that can be repurposed based on the provided gene signatures. The resulting drug candidates were ranked based on their Tau scores, and candidates with false discovery rates (FDR) of less than 0.05 were considered.

## 3. Results

### 3.1. Weighted Gene Co-Expression Network Analysis (WGCNA)

#### 3.1.1. Data Preparation and Scale-Free Networks

In WGCNA, the final gene expression matrices are prepared following a series of pre-processing steps, including normalization, outlier detection, and removal [17,19]. Outliers detected in the sample clustering dendrograms (Figure A1, Figure A2, Figure A3 and Figure A4) were filtered out. A total of 25,138 genes that were common across the datasets were utilized to construct the weighted gene co-expression networks. This process ensures the robustness and reliability of the network analysis, providing valuable insights into the intricate relationships among genes as it emphasizes networks with stronger correlations over weaker ones to capture complex interactions among genes [17,19]. Figure 1 presents the relationship between the scale-free topology fit index and soft threshold (β) values ranging from 1 to 20. As shown, the soft threshold at β > 5 starts to plateau and stabilize across all datasets, indicating that the scale-free topology fit is unaffected by increased power and results in a robust scale-free structure. Hence, it was decided to use a soft threshold power of β = 5 to approximate the scale-free topologies for all datasets. This decision holds significance to ensure the robustness and efficiency of the gene network, since selecting a lower soft-thresholding power results in denser gene networks with more connections. This approach increases the probability that the gene clusters will exhibit biologically relevant relationships between genes. As WGCNA suggests, ideally, the soft threshold power should have an *R*^2^ greater than 0.8, which can be visually observed in the scale-free topology fit as the point where datasets stabilize and plateau onwards. Thus, selecting a soft-threshold power of 5 would provide a more meaningful result [20,21,22].

With the chosen soft-thresholding power of β = 5, the soft connectivity (k) was calculated to assess the degree of fit of each dataset. This was performed by plotting the soft connectivity and calculating the log-log plot. Shown in Figure 2 is the corresponding log–log plot of the OC dataset, which represents the best fit among the dataset by having an *R*^2^ value of 0.96. This indicates that the OC dataset should be chosen as the reference dataset for WGCNA network construction and module identification, as the gene co-expression networks generated from the dataset would result in biologically relevant expression patterns [10]. That is based on the assumption that real biological networks tend to exhibit scale-free properties, and the dataset with the highest *R^2^* value increases its likelihood of representing functionally relevant biological networks amongst the datasets [23].

#### 3.1.2. Network Construction and Module Identification

In performing WGCNA, one technique is to designate one dataset as a reference for plotting the gene cluster dendrogram and generating modules. In this study, OC was chosen as the reference dataset, which transformed an adjacency matrix utilizing the selected soft threshold power of β = 5. To determine modules within the established network, a cluster dendrogram was generated using a weighted correlation matrix. Grouping genes into modules requires network proximity measurements, translating into similarities between genes within the dataset. Hence, TOM similarity measurement was performed to calculate the relativity between genes across the datasets, and the TOM dissimilarity index was used for plotting the gene dendrogram (Figure 3) that summarizes the identified modules. Overall, genes were segmented to form modules within the network, resulting in the identification of 33 modules using dynamic tree cut. These modules were further merged with similar expression profiles to yield more significant results. The modules in dynamic tree cut can be merged with similar expression profiles into more meaningful modules by calculating eigengene dissimilarity measurement and clustering its module dendrogram. Modules that had greater than 75% correlation value [24] were treated as highly related modules and, thus, will be merged, resulting from the original 33 modules from a dynamic tree cut to 23 eigengene modules. The identified modules were classified arbitrarily based on colors (Table A1): blue (1952), brown (2164), cyan (1594), darkgreen (181), darkgrey (116), darkorange (102), darkturquoise (143), magenta (984), midnightblue (1232), orange (107), paleturquoise (41), pink (1053), purple (947), royalblue (228), violet (35), yellow (1424), green (1630), saddlebrown (57), grey (327), skyblue (66), grey60 (354), lightcyan (425), and tan (935). These modules are gene clusters exhibiting similar expression patterns that can provide substantial information about the dysregulated processes or pathways involved in endometriosis and the gynecological cancers analyzed.

### 3.2. Module Preservation Analysis

In this network analysis, module preservation validates the robustness of identified modules across different gynecological disease datasets. It plays a pivotal role in assessing the reproducibility of co-expression networks and gauging their biological significance. Modules with a *Z_summary_* value exceeding 10 are deemed highly preserved [25]. Moreover, the gene count within each module serves as an indicator of its significance, with modules containing a larger number of genes being more meaningful as it implies more connectivity patterns [25]. The preservation of Z_summary_ values per dataset is shown in Figure 4, while the highly preserved modules identified across the datasets—cyan, midnightblue, pink, purple, and tan—are detailed in Table A2, Table A3 and Table A4. These findings underscore the robustness and significance of the identified modules in elucidating common molecular mechanisms underlying various gynecological diseases.

### 3.3. Gene Ontology (GO) and Kyoto Encyclopedia of Genes and Genomes (KEGG) Pathways

Following the conversion of Affymetrix gene IDs to Entrez IDs using the DAVID web server, the genes in highly preserved modules underwent functional annotation and pathway enrichment analysis to determine functional GO, focusing on BP, MF, CC, and KEGG pathway domains. The enriched GO and KEGG terms were examined for each module and compiled in Table A5, Table A6, Table A7, Table A8 and Table A9 while the top terms are shown in Figure 5. It can be observed that most of the modules primarily affect transcription regulation, as indicated by the midnightblue, cyan, tan, and purple modules, while the pink module was indicated to affect cell division. The highly preserved modules operate within the nucleus or cytosol, while all of them are associated with protein binding. The KEGG pathway analysis showed that the pink module affects the cell cycle, while the midnightblue module showed high similarity in pathways involved in Herpes Simplex Virus (HSV) infection, and cyan, tan, and purple modules were involved in transcription-related pathways algorithm (Appendix A).

### 3.4. Protein–Protein Interaction (PPI) Networks and Hub Genes

The STRING knowledgebase provided the protein-protein interaction (PPI) for the identified hub genes. The PPI network was specified to focus on protein interaction in humans. [26]. Through STRING, PPI networks were constructed to illustrate the gene interactions for each preserved module between the proteins of the corresponding genes in the highly preserved modules [26]. The threshold for the protein–protein interaction network usage was set at high confidence (>0.7). Each PPI network constructed was imported to Cytoscape, and the CytoHubba feature revealed the top 10 hub genes within each module of interest based on the degree centrality algorithm (Appendix A). Figure 6 shows the hub genes for each highly preserved module, where the genes in red color exhibit higher interaction scores. Since the identified hub genes exhibit high interaction scores within the PPI networks of their respective modules, further analysis would be beneficial in elucidating their potential roles in the key pathways and processes associated with gynecological diseases.

### 3.5. Signature-Based Approach for Drug Repurposing

The drug repurposing was based on the deregulated genes, differentially expressed genes (DEGs), revealed after the GEO2R in contrast with gynecological disease versus normal or control samples [27]. Those with fold-change (FC) of positive values were considered upregulated, while genes with negative values were downregulated. The transcriptional and molecular signatures of the upregulated and downregulated hub genes for each highly preserved module were analyzed through the DRE web server [https://www.drugrep.org/accessed on 16 April 2024]. Table 2 presents the top drug candidates along with their relevant mechanisms of action. Emphasis was given on drugs that exhibited more negative Tau values and lower FDR, as compounds with more negative Tau values induce a more countering effect on the genes provided, while those with low FDR are associated with fewer false positive drug discovery results. For the upregulated hub genes, thiamphenicol, trimethoprim, medrysone, pentolinium, and paroxetine were the top drug candidates, whereas propofol, fluconazole, dapsone, hydrocortisone, and MDL73005EF were the top drug candidates for the downregulated hub genes.

## 4. Discussion

### 4.1. Gene Co-Expression Modules across the Datasets

Early detection has been one of the leading causes of poor prognosis in patients suffering from gynecological cancers. This led to the development of several techniques for early detection and screening of gynecological cancers, such as the identification of biomarkers or liquid biopsy analyses [28,29,30]. Furthermore, a recent study has also discovered an increased risk for uterine sarcoma and endometrial cancer following endometriosis or pelvic inflammatory disease (PID) [31]. WGCNA analysis [32] was applied to scan for existing disease networks in gynecological cancers, represented by the OC, CC, and EC datasets with regard to the EM dataset. The functional annotation and pathway enrichment analysis through DAVID gene ontology has shown that most of the highly preserved modules, cyan, purple, midnightblue, and tan modules, were primarily involved with transcription dysfunction. The KEGG pathway analysis of these modules showed their involvement in spliceosome activity, actin cytoskeleton regulation, and Herpes simplex virus 1 infection. Given that these cellular events are associated with inflammation and immune response [11,12,33,34], it can be speculated that inflammation is a significant contributor to gynecological diseases. This can be further elaborated in Table 3, where several identified modules exhibited KEGG pathways that were found to be relevant in inflammation and immune response, including IL-17 signaling, Ras signaling, relaxin signaling, cytokine-cytokine receptor interaction, p53 signaling, and even viral carcinogenesis and HSV infection. The IL-17 signaling is a key pathway in inflammation as it promotes the expression of pro-inflammatory cytokines and chemokines and recruits immune cells to sites of inflammation [35]. On the other hand, Ras signaling is involved in the activation of inflammatory response [36], while relaxin signaling is involved in pathways that suppress inflammation [37]. Cytokine–cytokine receptor interaction refers to molecular pathways that describe the interaction between cytokines and cell surface receptors that regulate inflammation and immune response [38]. Lastly, viral infections among gynecological diseases have been known to cause chronic inflammation that could potentially cause carcinogenesis [39].

### 4.2. Module Hub Genes and Their Protein Functions

#### 4.2.1. Inflammatory Pathways Affecting Transcription Dysfunction

Among the hub genes identified (Figure 6), it can be observed that most hub genes with high interaction scores are involved in signaling pathways that are known to regulate inflammation. Such genes that are primarily involved in the NF-kB signaling pathway, whose primary function is to induce the expression of pro-inflammatory cytokines and chemokines, were included [48]. The upregulation of the *CASP3* gene indicates increased pyroptosis, which was found to be negatively correlated with the survival of cervical cancer patients [49], while the *SQSTM1* gene, known to increase inflammasome activity and oxidative stress-induced inflammation through p62, was also upregulated [48]. The downregulation of the *TP53* gene further shows impaired resolution of inflammation, resulting in the failure of the body to effectively halt the inflammatory response and restore tissue homeostasis [50]. The *STAT3* gene is a central player in inflammation through the JAK-STAT pathway. Its upregulation further activates the JAK-STAT signaling pathway, which is known to trigger cellular inflammation response, resulting in increased transcription and, eventually, carcinogenesis [51]. On the other hand, the upregulation of the *KRAS* gene indicates increased activation of the MAPK-ERK signaling pathway that is known to promote cytokine production, inflammasome activation, and amplification of inflammatory signals [52]. The results also indicate the upregulated activation of the mTOR signaling pathway through the upregulated *EGFR* gene. Overexpressed mTOR signaling is known to increase the production of pro-inflammatory cytokines [53]. All in all, these pathways are also known to regulate RNA splicing and processing [54,55,56,57], which further supports the involvement of inflammatory signals in transcription deregulation observed in gynecological cancer and condition datasets. Conversely, several genes that are involved in RNA splicing and processing, such as *HNRNPA1*, *HNRNPA2B1*, *RBM39*, and *SRSF11*, were also shown to have a potential impact on inflammation. A recent study showed that the knocked-down function of *HNRNPA1* and *HNRNPA2B1* genes resulted in decreased expression of IL-6 transcriptional activity [58]. Similarly, the knock-down of the *RBM39* gene showed reduced induction of interferon-stimulated genes [59].

#### 4.2.2. Viral Infections Triggering Inflammatory Pathways

In cervical cancer, HPV infection has long been known as a primary risk factor due to its disruption of the normal cellular regulatory mechanisms in cervical epithelial cells [60]. Aside from promoting cell proliferation, it often triggers the persistent release of pro-inflammatory cytokines and chemokines, leading to chronic inflammation. This creates a microenvironment that is prone to mutations and genomic instability, which facilitates the progression of precancerous lesions to invasive cervical carcinoma [7]. Among the highly preserved modules, the midnightblue module exhibited strong similarity with the pathways involved in HSV infection. Like in HPV, HSV infection also causes the release of pro-inflammatory cytokines and chemokines that lead to chronic inflammation, as well as tissue damage and ulceration at the site of infection that triggers immune response [40]. Several studies have indicated a higher risk of cervical cancer with HPV/HSV-2 coinfection [61]. Involved inflammatory pathways that are deregulated upon HPV and HSV infection, such as the NF-kB signaling, MAPK-ERK signaling, and JAK-STAT signaling pathways [40,62,63,64,65,66], were also included in the KEGG pathway results (Table 3). This further indicates the significance of these inflammatory pathways in gynecological disease prognosis.

### 4.3. Hormonal Imbalance and Inflammation

Estrogen, together with progesterone, are ovarian-derived steroid hormones that participate in facilitating the endometrium cycle in preparation for the menstruation cycle. However, overexpression of estrogen may occur during this cycle, which could potentially lead to progesterone resistance that may cause hormonal imbalance and inflammation sensitivity [67,68,69,70]. Progesterone resistance was thought to reduce cellular responsiveness to progesterone receptors, which may lead to gynecological diseases [67,71]. In endometriosis, debris from menstrual reflux containing endometrial glands and stroma in the fallopian tubes to the pelvic cavity causing lesions and/or extensive adhesion to lesions is its primary cause [67,71] High estrogen levels or overexpression may affect the ratio of estrogen receptor alpha and beta (ERα/ERβ) triggering an increase in pro-inflammatory cytokine IL-6 and overexpression of GREB-1 and c-Myc [72].

In the analysis conducted, the upregulation of the *ESR1* gene has been observed (Table 2). Furthermore, several hub genes (Figure 6) relating to signaling pathways that are associated with ERα overexpression were also involved, such as the MAPK-ERK, Ras, and JAK-STAT signaling pathways. This upregulation is thought to further amplify the inflammatory activity of these pathways [73,74,75]. Following the results in the PPI network construction and hub genes, *ESR1* upregulation has also been observed in the events of gynecological cancers, such as EC, OC, and CC, where a novel prognostic implication and probable specific endocrine therapy associated with *ESR1* expression [76]. High levels of estrogen-related receptors were formerly observed in ovarian cancer with poor overall survival in patients [77], while a GWAS study indicates a strong association between genetic risk variants on chromosome 6q25 and endometriosis, implicating the role of the *ESR1* gene and its correlated expression with related genes [78]. Similarly, *ESR1* mutations were observed in cervical cancer patients, which were thought to participate in the carcinogenesis of cervical squamous cell carcinoma [79]. The *ESR1* p.Y537S hotspot mutation was also observed in low-grade endometrial stromal sarcomas with histologic high-grade transformation and resistance to endocrine treatment [80]. Hence, this aligns with the results and may show a possible connection between *ESR1* mutations/ERα activation and signaling pathways in gynecological diseases.

Figure 7 shows the identified hub genes, signaling pathways, and biological processes that are potentially related to inflammation and ERα activation. *KRAS* gene mutations may lead to dysregulation in ERα activation that may lead to cancer proliferation, progression, tumor cell survival response, and metastasis [81,82]. In mice, *SOS1* knock-down suppresses the Ras-ERK/P38MAPK/JNK pathway and induces epithelial-mesenchymal transition through NF-kB signaling, which may contribute to the metastasis of epithelial ovarian cancer cells [83]. The *SOS1* gene is a co-repressor of ERα and coactivates *STAT3* [84], which may be influenced by the leptin-induced signal transduction that may result in transactivation of *STAT3* [85]. ERα activation may also affect the *EGFR* gene expression and may result in the upregulation of the mTOR/PI3K/AKT/MAPK signaling pathway and cause tumor growth [86]. *CTNNB1* gene mutations are associated with the recurrence of endometrial cancer and overall survival [87], possibly modulated by estrogen [88]. On the other hand, *TP53* hotspot mutations may lead to mutated p53 function and were found to be common among high-grade serous ovarian carcinoma undergoing primary treatment [89]. The *TP53* gene was also shown to have a strong hormonal influence through the ER signaling pathway, which may serve as a prognostic marker [90], as ERα activity could repress p53 transcription and potentially lead to decreased apoptosis and increased tumor cell proliferation [91,92]. This further supports the results, as the *TP53* gene was found to be downregulated in EM, OC, CC, and EC. In TGF-β/SMAD signaling, TGF-B1 inhibits the growth of normal ovarian surface epithelial cells and triggers the nuclear translocation of *SMAD4* [93,94]. ERα activation is known to affect the TGF-β signaling pathway, which may act as both a suppressive and promotive pathway for inflammation [95]. Similarly, the Wnt signaling pathway may also promote carcinogenesis, malignant transformation, and inflammation in cancer cells [96]. The PI3K-Akt signaling pathway also contains downstream protein targets, which may promote inflammation and are upregulated upon ERα activation [97]. Lastly, the transcriptional processes of RNA processing and splicing, which may both contribute to inflammation by promoting the expression of inflammatory genes and immune cell activity, were also found to be influenced by ERα activation [98].

### 4.4. Signature-Based Approach for Drug Repurposing

Several inflammatory pathways involved in transcription regulation, viral infection, and estrogen receptor activation have been highlighted (Table 3). Through the DRE web server, the top 5 hub genes of the upregulated and downregulated hub genes were identified. For the upregulated genes, the leading candidate drug is thiamphenicol, which is primarily known for inhibiting bacterial protein synthesis and used in treating PID. PID is a condition that has long been known to increase the risk of ovarian cancer as it induces sustained inflammation in the ovarian microenvironment. A recent study has reported that a history of PID increases the risk of having borderline ovarian tumors [99]. Hence, thiamphenicol could be a potential therapeutic option in ovarian cancer and other gynecological diseases due to its anti-inflammatory properties [100,101]. On the other hand, propofol was the leading drug candidate based on the downregulated hub genes. It is a benzodiazepine receptor agonist that is commonly used as an anesthetic agent [102], although previous studies have shown that the drug possesses anti-inflammatory effects that may play a role in modulating cancer progression and immune response [103]. This may be due to benzodiazepine derivatives that have been found to exhibit anti-inflammatory and analgesic tendencies [104].

Other drug candidates include trimethoprim, which inhibits the activity of the dihydrofolate reductase enzyme, which has long been a target for anti-inflammatory and anti-cancer treatment [105,106]. Another is a cholinergic receptor agonist, pentolinium, which is commonly used as an antihypertensive. Cholinergic receptors are known to activate anti-inflammatory pathways [107]. Fluconazole, a sterol demethylase inhibitor, is used to treat fungal or yeast infections such as vaginal candidiasis and various candida infections causing inflammation in other parts of the body [108]. On the other hand, dapsone is a bacterial antifolate that is commonly used to treat leprosy and dermatitis herpetiformis but also has known anti-inflammatory effects due to inhibiting reactive oxygen species production [109]. Furthermore, glucocorticoid receptor agonists, such as medrysone and hydrocortisone, were also included. This may be due to glucocorticoid receptor activation repressing gene expression of several biological processes, including inflammation [110]. Lastly, paroxetine, a selective serotonin reuptake inhibitor, and an experimental serotonin receptor antagonist drug, MDL73005EF, were also included. The role of serotonin as a chemotactic agent in inflammation, by increasing pro-inflammatory cytokine secretion, has been studied in the past years [111]. Overall, the identified drug candidates are known to have anti-inflammatory effects that could potentially inhibit the progression of gynecological diseases, as suggested in the analysis conducted. Studies have also indicated that anti-inflammatory drugs are beneficial in decreasing menstrual bleeding, postoperative treatment, and pain management in endometriosis, further supporting these results [112,113].

## 5. Conclusions

In this study, five highly preserved modules in different estrogen-dependent gynecological diseases that were represented by EM (GSE51981), OC (GSE63885), CC (GSE63514), and EC (GSE17025) datasets were successfully determined. Through WGCNA, the top 10 hub genes per module were identified and used to screen for potential drug candidates. Moreover, the functional annotation and pathway enrichment analysis allowed for the elucidation of the complex molecular basis for endometriosis and gynecological diseases, which highlighted several inflammatory pathways that are linked with transcription regulation. Based on the hub genes, signaling pathways associated with viral infections, particularly HSV, associated with gynecological cancers were also identified. The involvement of the *ESR1* gene, which encodes for ERα, a known modulator of signaling pathways involved in inflammation, further indicates its importance in gynecological disease prognosis. The drugs thiamphenicol, a bacterial ribosomal subunit inhibitor, and propofol, a benzodiazepine receptor agonist, were the top candidates identified in the signature-based drug repurposing. Other drug candidates include a dihydrofolate reductase inhibitor, glucocorticoid receptor agonists, cholinergic receptor agonists, selective serotonin reuptake inhibitors, sterol demethylase inhibitors, bacterial antifolate, and serotonin receptor antagonist drugs which have known anti-inflammatory effects, demonstrating that the gene network can be used as a therapeutic avenue to design drug candidates for gynecological diseases. However, further evaluation through wet laboratories is required as the data presented were conducted in silico.

## Figures and Tables

**Figure 1 biology-13-00397-f001:**
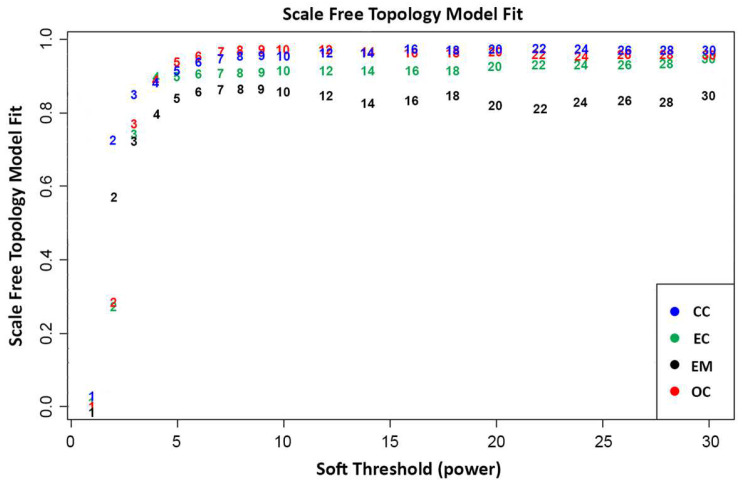
Scale-free topology model fit plot of network indices calculated through the average number of connections per gene in the networks. This indicates the scale-free topology model fit per dataset at a given soft thresholding power.

**Figure 2 biology-13-00397-f002:**
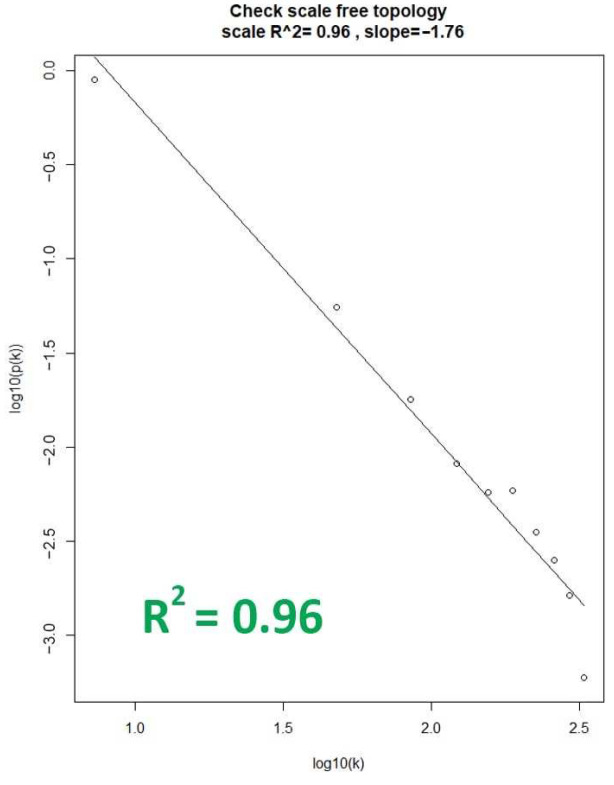
Linear relationship plot of the OC dataset. The X-axis represents the degree (k) of a node in the network, while the y-axis represents the probability (p(k)) of finding a node with a given degree.

**Figure 3 biology-13-00397-f003:**
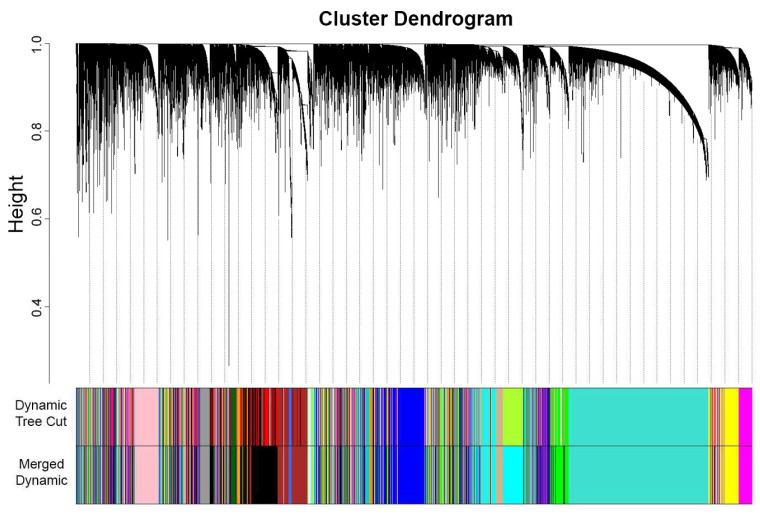
OC gene clustering dendrogram of TOM dissimilarity of 33 modules from the dynamic tree cut were merged into 23 modules. The continuous decrease in height (curving) displays similarity in gene expression, thus forming modules that are represented arbitrarily by colors.

**Figure 4 biology-13-00397-f004:**
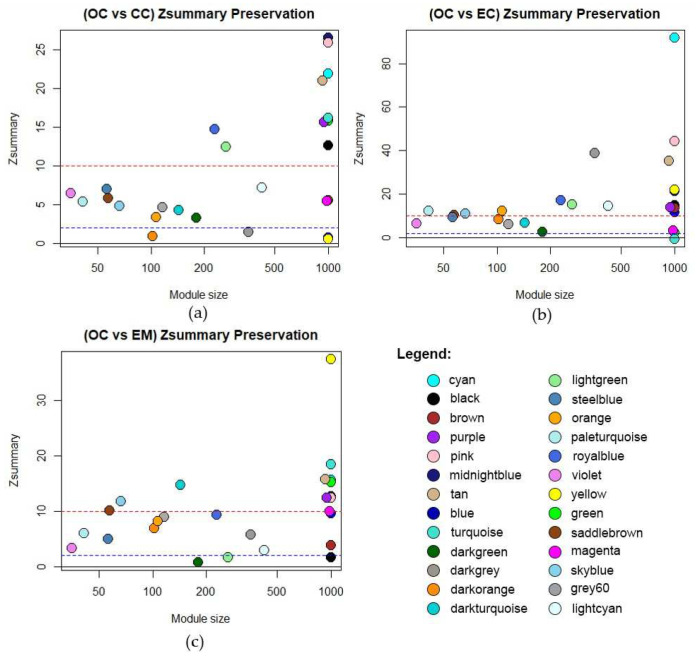
Module preservation of identified modules from OC network compared to (**a**) Cervical Cancer, (**b**) Endometrial Cancer, and (**c**) Endometriosis. The dashed red line at Z_summary_ > 10 implies strong evidence of preservation. Between the dashed blue and red line at 10 > Z_summary_ > 2 implies moderate evidence of preservation. Below the dashed blue line, Z_summary_ < 2 implies weak evidence that the module is preserved.

**Figure 5 biology-13-00397-f005:**
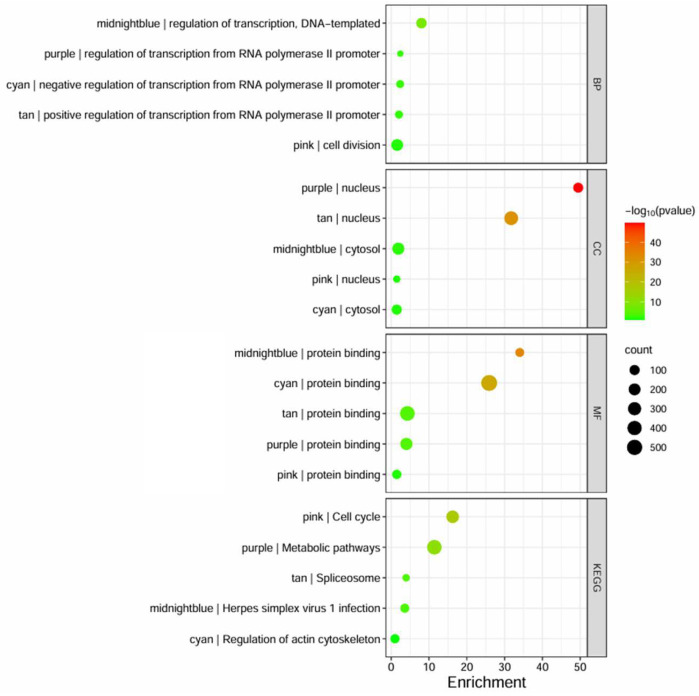
The bubble plot represents the GO terms and KEGG enrichment pathways, based on DAVID, that are highly enriched for each identified preserved module. The size of the bubble indicates the number of genes involved, the color intensity (from red to green) represents the *p*-value score, and the x-axis represents the enrichment fold score.

**Figure 6 biology-13-00397-f006:**
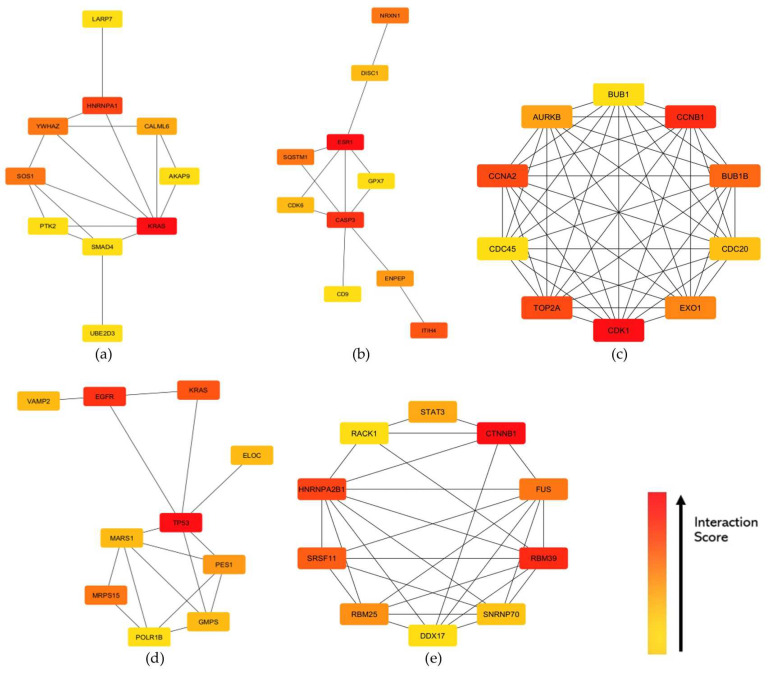
Top 10 hub genes identified based on degree centrality in the constructed PPI networks of the (**a**) cyan, (**b**) midnightblue, (**c**) pink, (**d**) purple, and (**e**) tan modules. Interaction scores were indicated by the intensity color of red gradient to yellow, where the red node has the highest interaction score across the network while yellow has the least.

**Figure 7 biology-13-00397-f007:**
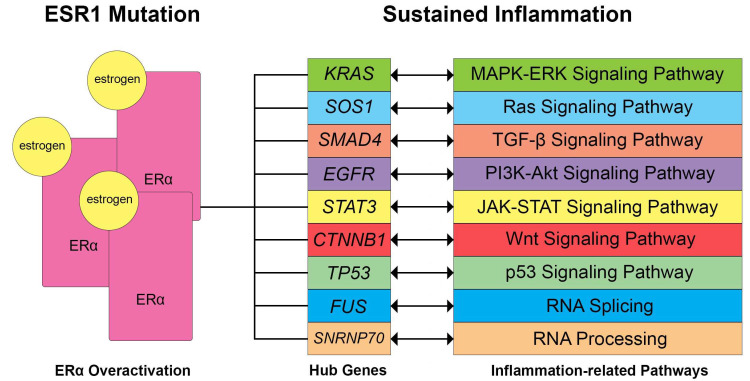
Potential dysregulated signaling pathways and biological processes related to sustained inflammation through ERα overexpression based on literature. *ESR1* mutations observed in gynecological diseases usually lead to increased expression of ERα, thereby influencing the hub genes and affecting the following signaling pathways. Double-ended arrows signify that ERα activation may induce regulatory effects on the gene expression, affecting signaling pathways or vice versa.

**Table 1 biology-13-00397-t001:** GEO Datasets Summary.

Accession No.	GSE51981	GSE63885	GSE63514	GSE17025
Condition	Endometriosis	Ovarian Cancer	Cervical Cancer	Endometrial Cancer
Year Published	2014	2014	2015	2011
Type	Expression Profiling by Array			
Platform	GPL570-HG-U133 Plus 2 Affymetrix Human Genome U133 Plus 2.0 Array			
Source	Endometrial Tissue	Primary Cancer Tissue		
No. of Samples	77	101	104	91

**Table 2 biology-13-00397-t002:** Top drug candidates based on the upregulated and downregulated hub genes.

Expression	Genes	Drug	Mechanism	Tau	FDR
Upregulated	*KRAS*, *HNRNPA1*, *SOS1*, *YWHAZ*, *AKAP6*, *LARP7*, *PTK2*, *UBE2D3*, *SMAD4*, *CASP3*, *SQSTM1*, *ESR1*, *NRXN1*, *ENPEP*, *GPX7*, *CDK1*, *CCNB1*, *TOP2A*, *CCNA2*, *BUB1B*, *EXO1*, *AURKB*, *CDC20*, *BUB1*, *CDC45*, *EGFR*, *MRPS15*, *GMPS*, *MARS1*, *ELOC*, *POLR1B*, *CTNNB1*, *RBM39*, *HNRNPA2B1*, *RBM25*, *DDX17*, *SRSF11*, *RACK1*, and *STAT3*	Thiamphenicol	bacterial 50S ribosomal subunit inhibitor	−99.8	5.24 × 10^−3^
		Trimethoprim	dihydrofolate reductase inhibitor	−99.7	9.69 × 10^−3^
		Medrysone	glucocorticoid receptor agonist	−99.1	5.79 × 10^−3^
		Pentolinium	cholinergic receptor agonist	−98.9	4.72 × 10^−4^
		Paroxetine	selective serotonin reuptake inhibitor	−98.8	1.50× 10^−3^
Downregulated	*CALML6*, *ITIH4*, *CDK6*, *DISC1*, *CD9*, *TP53*, *PES1*, *VAMP2*, *FUS*, *and SNRNP70*	Propofol	benzodiazepine receptor agonist	−99.5	2.11 × 10^−3^
		Fluconazole	sterol demethylase inhibitor	−99.4	1.74 × 10^−3^
		Dapsone	bacterial antifolate	−99.1	8.96 × 10^−3^
		Hydrocortisone	glucocorticoid receptor agonist	−98.8	8.22 × 10^−3^
		MDL73005EF	serotonin receptor antagonist	−98.7	4.68 × 10^−3^

**Table 3 biology-13-00397-t003:** Summary of KEGG pathways related to inflammation and immune response in identified modules.

Module/s	KEGG Pathway	*p*-Value	Reference
Cyan	Ras signaling pathway	1.0 × 10^−1^	[36]
Midnightblue, Green	Herpes simplex virus infection	3.6 × 10^−7^	[40]
Pink	p53 signaling pathway	9.1 × 10^−5^	[41]
Purple	Apelin signaling pathway	3.0 × 10^−2^	[42]
Tan	Spliceosome	1.2 × 10^−4^	[43]
Brown	Cytokine-cytokine receptor interaction	5.3 × 10^−8^	[44]
Darkgrey	Relaxin signaling pathway	1.01 × 10^−1^	[37]
Lightcyan	Antigen processing and presentation	9.5 × 10^−23^	[45]
Magenta, Skyblue	MAPK signaling pathway	2.1 × 10^−3^	[46]
Orange	IL-17 signaling pathway	4.0 × 10^−15^	[35]
Steelblue	Viral carcinogenesis	1.1 × 10^−18^	[47]

## Data Availability

The gene microarray datasets used for the study are openly available in the NCBI Gene Expression Omnibus (GEO) database under the accession IDs GSE51981, GSE63885, GSE63514, and GSE17025 datasets at https://www.ncbi.nlm.nih.gov/geo/ accessed on 30 October 2023.

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
