# Peer review of "Transcriptomic Analysis of Hub Genes Reveals Associated Inflammatory Pathways in Estrogen-Dependent Gynecological Diseases"

_biology, 2024, doi:10.3390/biology13060397_

Round 1

Reviewer 1 Report

Comments and Suggestions for Authors

Elaine C. Pasamba et.al investigated molecular mechanisms related to gynecological diseases by WGCNA analysis using 4 published microarray datasets. They have identified preserved gene modules and hub genes, which are potential candidates for drug discovery. Through GO, KEGG, PPI network and Hub gene analysis,  the author found the key pathways and discussed their functions in gynecological diseases. The authors also focus on the differential regulated hub genes, they searched drug candidates for these hub genes. Overall, this paper is well-written and results and figures clearly illustrated, Reading through the manuscript, I have a few comments on it.

1.        The figure legend in Figure 6 is missing, the author should provide the color scale of the network.

2.        In Table 3, the author lists the KEGG pathways related to inflammation and immune response, are these pathways enriched in those modules? The author should provide the enriched p-value for these pathways.

3.        The author discussed the potential deregulated signaling pathways and biological processes related to inflammation in ERα overactivation. However, it is still not clear which hub genes in the analysis are related to inflammation, I suggest the authors give some hub gene examples they found in the main figures.

Author Response

Dear Reviewer, 

Good day! Thank you for your valuable feedback on our manuscript. We sincerely appreciate the time and effort you've put into providing such a thoughtful review. Your comments have contributed to enhancing the quality of our paper. Please see the attachment.

Sincerely yours,

Elaine C. Pasamba

Reviewer 2 Report

Comments and Suggestions for Authors

In the Manuscript titled “Transcriptomic Analysis of Hub Genes Reveals Dysregulated 2 Inflammatory Pathways in Estrogen-Dependent Gynecological 3 Diseases”, the Authors analyze 373 microarray samples containing transcriptomic information on endometriosis, as well as ovarian, cervical, and endometrial cancers. Following RMA normalization, network construction and module building using the WCGNA protocol, thirty-three preserved gene co-expression modules were identified based on adjacency data relative to the reference ovarian cancer dataset. The modules were further validated using Module Preservation Analysis resulting in twenty-six highly preserved gene co-expression modules. For each module GO and KEGG pathway analyses were performed. Then, using STRING and Cytoscape, PPI networks and hub genes with maximum connectivity were identified. Next, gene signatures for statistically significantly up- and down-regulated hub genes were used to scan MSigDB and CMap databases for expression signatures of drugs that could be re-purposed for the above diseases resulting in identification of ten drugs belonging to a variety of pharmacological classes, five each for the up- and down-regulated gene expression signatures for the hub genes. Finally, and unexpectedly, the Authors present a hypothesis to explain the data based on ER-α activation and dysregulated inflammation. Using inflammation as a common denominator, the Authors propose a common set of pathways that could be involved in endometriosis, ovarian, cervical, and endometrial cancers and that could be targeted by the repurposed drugs.

General comments: The above pipeline for microarray data extraction, normalization, co-expressed gene pathways construction, hub gene extraction, and signature-based repurposed drugs builds on a robust set of bioinformatics tools previously validated by published studies. The Authors appear to be well familiar with these protocols, describe them competently in Methods and provide relevant and current references throughout the Manuscript.

One overall criticism regarding methodology is that each step relies on specific assumptions that need to be scrutinized and understood in terms of their strengths and limitations. For example, STRING is a compilation of protein-protein interaction networks extracted from various in vitro and in vivo, vertebrate as well as invertebrate models that may not provide an appropriate representation of human diseases. Also, targeting down-regulated genes with repurposed drugs may be an appealing computational model but more difficult to translate therapeutically. 

Unfortunately, even as these limitations are not addressed, the Discussion section overreaches beyond the data and becomes disjointed, raising concerns regarding critical thinking and data interpretation.

Major concerns: The Discussion is long and speculative trying to connect high ESR1 to inflammatory signals and cancer.

Having found that ESR1 is upregulated (the reader assumes this but data on up-and down-regulated genes with FDR and statistical significance is missing) the Manuscript tries to establish a connection with “dysregulated inflammation” presumably driven by ESR1. This connection is contradicted by common sense evidence. Estrogen levels in women peak between the ages of 20 and 30 years of age and then return to levels approximately one-tenth by age 50. By contrast, gynecological cancers are lower in the 20-30 years of age bracket and increase almost linearly with age after 50. So, when ESR1 levels are highest the incidence of gynecological cancers is lowest and when ESR1 levels are lowest, incidence of gynecological cancers is highest.

Similarly, if a connection existed between ER-α and “dysregulated inflammation” that would be apparent during the oestrus cycle. A spike in inflammation during the follicular phase of the oestrus cycle when ER-α levels are highest would be expected together with a corresponding decline during the progesterone phase when ER-α levels decline.

Having tried to establish a non-existent connection between high ESR1 and “dysregulated inflammation”, the Manuscript then defers to the inflammation-cancer paradigm to explain findings in a way that is beyond the scope of the study, and which further undermines its findings. This is primarily evidenced by the fact that classical anti-inflammatory drugs are not picked-out with the up-regulated gene signature.

Rather than build on the novelty of findings which follow the well-established discovery pipeline, Figures 7 and 8 create confusion rather than clarity and produce a self-conflicting Manuscript.

Minor concerns: Issues with what the readers may perceive as inappropriate comments related to sexual behavior (likely unintended but poorly phrased), poor data labelling for Figs. 4-5, other identified in the attached Manuscript review.

Please check the attached file for more details.

Author Response

(The authors gave the same response as above.)

Reviewer 3 Report

Comments and Suggestions for Authors

The research conducted an analysis of gene expression focusing on EM, OC, CC, and EC, followed by the identification of gene modules. Subsequently, downstream biomarker and pathway-based analyses were performed to elucidate disease mechanisms and propose potential candidate drugs. The introduction effectively establishes the study's hypothesis, while the methodology adeptly outlines the utilization of methods such as WGCNA for network-related analysis. The discussion provides a thorough interpretation of the study's findings. However, there are some concerns:

  1. Regarding lines 115-117: When referring to genes without expression values, do you mean genes with a value of 0? Could you justify the exclusion of genes with zero values, considering their potential importance, especially in pattern recognition?

  2. In the methodology section: Could you specify the final gene expression matrices (for each case) utilized for WGCNA analysis? Please include this information.

  3. What was the threshold for PPI used in the STRING database?

  4. Figure 6 depicts nodes in different colors. Could you clarify the significance of each color?

Comments on the Quality of English Language

Minor editing is required in my opinion. 

Author Response

(The authors gave the same response as above.)

Round 2

Reviewer 2 Report

Comments and Suggestions for Authors

The Authors have now presented a revised version of the Manuscript that successfully addresses the issues raised in the initial review. No further criticism.

Author Response

Dear Reviewer,

Thank you very much for your time and effort.

Sincerely yours,

Elaine C. Pasamba

Reviewer 3 Report

Comments and Suggestions for Authors

All comments were addressed. Thanks for the revision. 

Author Response

(The authors gave the same response as above.)
